# A Patient Self-Made Point-of-Care Fecal Test Improves Diagnostic Accuracy Compared with Fecal Calprotectin Alone in Inflammatory Bowel Disease Patients

**DOI:** 10.3390/diagnostics11122323

**Published:** 2021-12-10

**Authors:** Gonzalo Hijos-Mallada, Raul Velamazán, Raúl Marti, Eduardo Chueca, Samantha Arechavaleta, Alberto Lué, Fernando Gomollón, Angel Lanas, Carlos Sostres

**Affiliations:** 1Department of Gastroenterology, University Hospital “Lozano Blesa”, 50009 Zaragoza, Spain; raulvs92@gmail.com (R.V.); alberto.lue@hotmail.com (A.L.); fgomollon@gmail.com (F.G.); angel.lanas@gmail.com (A.L.); carlossostres@gmail.com (C.S.); 2Instituto de Investigación Sanitaria Aragón (IIS Aragón), 50009 Zaragoza, Spain; educhuec@gmail.com (E.C.); samantha.arechavaleta@gmail.com (S.A.); 3Department of Medicine, University of Zaragoza, 50009 Zaragoza, Spain; raulmm94@hotmail.com; 4CIBERehd, 28029 Madrid, Spain

**Keywords:** fecal immunochemical test, fecal calprotectin, fecal lactoferrin, fecal transferrin, rapid fecal tests, ulcerative colitis, Crohn’s disease

## Abstract

Background: Monitoring inflammatory bowel disease patients may be challenging. Fecal calprotectin is one of the most performed tests. Other fecal biomarkers are less used in clinical practice. Rapid fecal tests that could be performed by patients may be a useful strategy to closely monitor disease activity. Methods: We performed a prospective observational study including consecutive inflammatory bowel disease patients referred for colonoscopy in a single center. Certest FOB + Transferrin + Calprotectin + Lactoferrin^®^ (Certest Biotec S.L, Zaragoza, Spain), a one-step point-of-care test which simultaneously detects these four biomarkers was performed. Endoscopic inflammatory activity was defined using the Mayo score (≥1) in ulcerative colitis, SES-CD (>3) and Rutgeerts scores (≥1) for Crohn’s disease. Results: Out of a total of 106 patients (56.5% female, mean age 51 years), 54 (50.9%) were diagnosed with ulcerative colitis and 52 (49.1%) with Crohn’s disease. Endoscopic activity was detected in 42 patients (39.0%). Fecal calprotectin provided the best sensitivity (97.6%), with limited specificity (34.4%). Compared to calprotectin, the other 3 fecal biomarkers showed better specificity (87.5–92.1%) and lower sensitivity (45.2–59.5%). Patients with a negative result in all biomarkers (19/106—17.9%) had 100% (CI 95% 97.4–100) negative predictive value, while patients with the 4 biomarkers positive (13/106—12.3%) had 100% (CI 95% 96.1–100) positive predictive value of endoscopic inflammatory activity. AUROC of this 4 biomarker point-of-care test was 0.845 (95% CI 0.771–0.920), significantly higher than the AUROCs of any of the 4 biomarkers. Conclusions: This test may be a useful strategy to monitor inflammatory activity in clinical practice by excluding or prioritizing patients in need of a colonoscopy.

## 1. Introduction

The incidences of both ulcerative colitis (UC) and Crohn’s disease (CD) are increasing worldwide [1,2,3]. Some aspects make the assessment of patients with IBD difficult. Firstly, UC and CD are chronic and progressive diseases that show different patterns, with marked heterogeneity in inflammatory activity as well as symptomatic burden and response to treatments, between patients and within the same patient over time [4]. Secondly, IBD patients may present with a wide variety of symptoms, which do not correlate well with inflammatory activity [5]. Finally, the non-invasive surrogate markers available do not have an exact correlation with gastrointestinal inflammation. As a result, in a relevant percentage of symptomatic IBD patients, performing a colonoscopy is necessary as mucosal healing has been associated with improved clinical outcomes (long-term remission, avoidance of surgery, and reduction in hospitalization) [6]. However, endoscopy is invasive, costly, and sometimes poorly accepted by patients, so reliable noninvasive biomarkers are necessary for disease monitoring.

Concerning fecal biomarkers, fecal calprotectin is an antimicrobial protein derived predominantly from neutrophils. It is a useful surrogate marker of gastrointestinal inflammation, significantly validated to discriminate functional from organic gastrointestinal disease [7,8]. Fecal calprotectin levels have been found to correlate with clinical, endoscopic and histological activity in UC [9] and CD [10], so currently it is the most commonly used fecal biomarker in this setting [11]. Nevertheless, despite the widespread availability of fecal calprotectin in the United States, only 2–5% of patients with UC undergo testing in routine practice [12].

Fecal occult blood tests are widely used for diagnosing colorectal cancer and precancerous lesions, in both population-based screening programs in average-risk population and in symptomatic patients [13]. Its diagnostic yield for detecting inflammatory activity in IBD is less established, although some studies have shown better outputs when calprotectin is associated with fecal hemoglobin [14,15,16]. Another marker of occult gastrointestinal bleeding is fecal transferrin, which is believed to be more stable in feces than hemoglobin. Therefore, it has been suggested that it may be a more sensitive marker, especially for lesions located in the proximal colon [17]. Its role in IBD monitoring has not yet been studied. Fecal lactoferrin, similar to calprotectin, has mainly been used to evaluate the inflammatory activity of IBD patients [18]. It is less used in clinical practice owing to its lower sensitivity and scarce evidence compared with fecal calprotectin [19].

In the last few years, the patient’s convenience has become increasingly relevant. Because of that, rapid point-of-care fecal tests based on noninvasive biomarkers have been emerging [16,20]. Due to their simple use and interpretation, these tests may be used in outpatient clinics, giving immediate results, or even by patients themselves. This could be an even more useful approach in present times, as minimizing hospital visits and endoscopic examinations are recommended strategies in the current COVID-19 pandemic. Studies show different levels of acceptance of these tests [21,22].

There is scarce evidence in the literature about the diagnostic accuracy of rapid fecal tests combining different biomarkers in IBD patients. In this study, we analyze the diagnostic accuracy of “FOB + Transferrin + Calprotectin + Lactoferrin^®^ (Certest Biotec S.L, Zaragoza, Spain)”, a one-step combo card fecal test for the simultaneous semi-qualitative detection of haemoglobin, transferrin, calprotectin and lactoferrin in fecal samples of IBD patients.

## 2. Materials and Methods

### 2.1. Study Population

We performed a single-center, prospective observational study, enrolling consecutive and independent IBD patients who underwent a follow-up colonoscopy from the health area of University Hospital “Lozano Blesa” (Zaragoza, Spain) between March and November 2019.

The patients were contacted one week before colonoscopy to be informed about the study. If they agreed to participate, they were asked to collect a stool sample 24 h before starting colonic preparation, to keep it refrigerated, and to bring it to the hospital at the date of the colonoscopy.

Patients older than 90 or younger than 18 who brought an invalid stool sample or presented inadequate colon preparation, which made ruling out the presence of endoscopic activity unfeasible were excluded from the final analysis.

### 2.2. Fecal Tests

The test that we performed on the stool samples was “FOB + Transferrin + Calprotectin + Lactoferrin^®^ (Certest Biotec S.L, Zaragoza, Spain)” a one-step coloured chromatographic immunoassay for the simultaneous semi-qualitative detection of human haemoglobin (Hb), human transferrin (Tf), human calprotectin (Cp) and human lactoferrin (Lf). The cut-off values of the test are 5.1 μg/g for Hb, 0.4 μg/g for Tf, 50 μg/g for Cp and 10 μg/g for Lf. This test is easy to perform. In brief, the stick of the test kit is used to take a sufficient fecal sample, which is mixed with a diluent in the collection tube. Following this procedure, four drops of the solution should be dropped in a circular window for each biomarker in the test cartridge, and lateral migration of the sample leads to formation of control and test lines. The result is provided in 10 min. The test was performed and read by trained investigators, blinded to clinical information and the colonoscopy results of the patients

### 2.3. Colonoscopy and Definitions

We defined the inflammatory activity according to validated endoscopic scores. For UC we used the endoscopic Mayo score [23]. We considered endoscopic activity grades 1–3 (Mayo 1: mild activity, Mayo 2: moderate activity, Mayo 3: severe activity), and absence of endoscopic activity grade 0. For patients with CD, we used SES-CD [24] considering endoscopic activity having 3 or more points (3–8 points: mild activity, 9–12 points: moderate activity, >12 points: severe activity) and the absence of endoscopic activity having less than 3 points. Finally, for patients with CD with previous surgery, we used the Rutgeerts score [25] considering activity grade 1–4 and the absence of endoscopic activity grade 0.

### 2.4. Statistics

A descriptive analysis of the patients included was performed. The Kolmogorov–Smirnov test was used to assess if continuous variables followed a normal distribution. Continuous variables were presented as mean with standard deviation or median with interquartile range. Qualitative variables were described with frequencies and percentages. Chi-square was used to evaluate the relationship between qualitative variables.

Sensitivity, specificity, negative predictive value (NPV), positive predictive value (PPV) and area under receiver operator curve (AUROC) were calculated with confidence interval 95% for each biomarker, for the diagnosis of endoscopic inflammatory activity. These figures were calculated for the whole cohort and also stratified by the diagnosis (UC and CD). The method of DeLong et al. [26] was used to assess the statistical significance of the differences between AUROCs. SPSS version 26 (IBM Corporation^®^), MedCalc version 13.3 (MedCalc Software^®^) and EPIDAT 3.1^®^ were used for statistical analysis

### 2.5. Ethics

The study design was reviewed and approved by the local ethics committee (CEICA—Regional Ethical Committee of Aragón—PI17/0172, approved on 10 May 2017). All subjects gave informed written consent preceding their participation in the study.

## 3. Results

### 3.1. Baseline Characteristics

A total of 125 patients were contacted, of whom 117 agreed to participate in the study (93.6% participation rate). Eleven patients were excluded due to exclusion criteria (4 due to insufficient bowel preparation and 7 because of an invalid stool sample). Thus, 106 participants were included in the final analysis. Population baseline characteristics are shown in Table 1.

### 3.2. Colonoscopy Findings and Fecal Test

Endoscopic inflammatory activity was detected in 42 (39.0%) patients: 17 (16.0%) had mild activity, 20 (18.0%) moderate activity and 5 (5.0%) patients had severe activity. Mucosal healing was identified in 64 (61.0%) patients.

Regarding UC population, 15 (27.7%) patients presented endoscopic activity: 6 (11.1%) had mild activity, 8 (14.8%) moderate activity and finally 1 (1.8%) had severe activity. On the other hand, 27 (51.9%) CD patients presented activity in the endoscopic evaluation, 11 (21.2%) mild activity, 12 (23.0%) moderate activity and 4 (7.6%) severe activity. Significantly more patients with CD presented endoscopic activity (*p* = 0.011).

The positivity rates of the 4 biomarkers were 78.3% for Cp, 26.4% for Lf, 31.1% for Hb and 25.4% for Tf. A total of 17.9% (19/106) of patients had a negative result in the four tests, whereas 12.3% (13/106) had positive results in the four biomarkers.

### 3.3. Diagnostic Accuracy of Fecal Tests

Table 2 summarizes the results and diagnostic accuracy of each fecal biomarker in all the patients included in the study. These results, stratified by disease, are shown in Table 3.

Table 4 summarizes the diagnostic accuracy of the combination of the 4 biomarkers according to the number of positive results. Table 5 shows this information stratified by diagnosis.

Figure 1 shows the distribution of patients with or without endoscopic activity regarding the number of positive tests of the combined rapid test.

The AUROC for the diagnosis of inflammatory activity of the combination of the 4 fecal biomarkers in the whole cohort was 0.845 (95%CI 0.771–0.920). The AUROC for the diagnosis of inflammatory activity in the UC subgroup was 0.868 (95%CI 0.756–0.979), while in the CD subgroup it was 0.841 (95%CI 0.737–0.945). No significant differences were found comparing the AUROCs of the combo test between both diagnoses (*p* = 0.7). The AUROC of the combination of the 4 fecal biomarkers for the diagnosis of endoscopic inflammatory activity is represented in Figure 2.

The AUROCs for the diagnosis of inflammatory activity of each biomarker in the whole cohort were: Cp 0.660 (95%CI 0.596–0.723), Lf 0.734 (95%CI 0.651–0.818), Hb 0.735 (95%CI 0.649–0.820), and Tf 0.664 (0.577–0.750). The AUROC of the combined rapid test was significantly higher than the AUROCs of any of the four biomarkers (*p* < 0.001). However, no significant differences were found when comparing the AUROCs of each biomarker individually.

In addition, we analyzed the relationship between the number of positive biomarkers and the severity of the endoscopic activity. Analyzing all the patients included in the study, we found significant association (*p* < 0.01) between the number of positive tests and the degree of activity detected. However, taking into account only the patients diagnosed with endoscopic activity (*n* = 42), the number of positive biomarkers was not related to the severity of endoscopic activity (*p* = 0.94).

## 4. Discussion

IBD monitoring may be challenging due to its irregular clinical course and the occasional presence of functional symptoms. Although colonoscopy continues to be the gold standard for the evaluation of mucosal healing, it is sometimes poorly tolerated by IBD patients, and carries a non-negligible risk of complications as well as an economic burden. In addition, the current COVID-19 pandemic has complicated the monitoring of all chronic diseases, including IBD patients. In a great number of European hospitals, avoiding or reducing the number of hospital visits has become one of the main strategies to minimize the risk of infection in most immunodeficient IBD patients [27]. Telemedicine has gained importance in this issue and has probably come to stay in the next few years. In this context, easy non-invasive tests like the one explored in our study, which can be performed by patients themselves, could be a useful strategy.

In this study, we have evaluated the association between 4 fecal biomarkers and the direct assessment of mucosal inflammation by colonoscopy in a cohort of IBD patients.

Regarding the diagnostic yield of each biomarker, Cp provided the best sensitivity (97.6%) and NPV (95.6%) outcomes. These figures are similar to the results of other studies with the same cut-off point of 50 µg/g, and offer a higher sensitivity than that reported in other studies using thresholds of 100–250 µg/g [7,10]. Given its high sensitivity and NPV, Cp performed well as a screening biomarker for endoscopic activity in our cohort, but it yielded low PPV (49.4%) and specificity (34.4%) (Table 2). Only one patient with negative Cp showed endoscopic activity, namely a UC patient with endoscopic Mayo 2, and 2 positive fecal tests (transferrin and hemoglobin), so probably microscopic mucosal bleeding was predominant. However, it should be noted that the AUROC of the Cp test was only 0.66, far worse than the data reported in most studies analyzing diagnostic accuracy of fecal calprotectin in IBD patients. This is probably related to the fact that the majority of calprotectin assays analyzed in these patients were quantitative ELISA tests, so sensitivity and specificity varied depending on the cut-off chosen [7]. Evidence regarding diagnostic accuracy of rapid qualitative fecal tests is less available, but one study with a similar design reported and AUROC of 0.75 using a rapid semi-quantitative calprotectin assay with three possible cut-offs (50 µg/g, 50–200 µg/g and >200 µg/g) [20]. With a cut-off of 50 µg/g, this rapid calprotectin test showed a sensitivity of 93% and specificity of 34%, almost identical to the results of our Cp test [28]. Concerning the other fecal biomarkers, Lf, Hb and Tf showed worse sensitivity and NPV, but provided better specificity (92.2%, 87.5%, 87.5%) and PPV (82.1%, 75.8%, 70.4%) compared to Cp (Table 2). A potential improvement in the diagnostic accuracy of a combination of these markers with Cp due to their higher specificity has already been suggested by previous studies [14,15,16,19].

In the subpopulation analysis, Cp had similar results in both UC and CD while Lf, as has been reported before [19], got slightly better results in the UC subpopulation (Table 3). Analyzing the combination of these 2 inflammatory biomarkers (Cp + Lf) in the whole cohort, 23 patients had negative results in both biomarkers and only one was diagnosed with inflammatory activity (NPV 95.6%). Both tests were positive in 28 patients, 5 of them with evidence of mucosal healing (PPV 82.1%).

Regarding the combination of the 4 biomarkers, the extreme test results (having all or no positive tests) had the best statistical outcomes. In case of 0 positive tests, the NPV was 100%, as 19 patients obtained that result and all of them were in endoscopic remission. On the other hand, 13 patients in our study received positive results in the four fecal tests and all of them presented endoscopic activity, which means a 100% PPV. When analyzing the intermediate results, considering the four possible cut-offs of the combined test according to the number of positive biomarkers, sensitivity and NPV decreased in parallel with the number of positive tests; whereas specificity and PPV increased in a similar way (Table 4).

These results would lead to reconsider performing the endoscopic evaluation in up to 19 patients who obtained 0 positive biomarkers, which would mean avoiding 17.9% (19/106) of all colonoscopies. In the remaining patients, the rapid fecal test could be useful to prioritize the need for a colonoscopy, taking into account the number of positive biomarkers. Those with 4 positive biomarkers (12.2%—13/106) could have been prioritized as all of them were later diagnosed with endoscopic activity. In patients with intermediate results (1–3 positive biomarkers) the rapid fecal test could be a useful tool to integrate into daily clinical practice to help the practitioner to prioritize the need for endoscopic evaluation or step up therapy, taking into account other parameters used in clinical practice [8,16].

Finally, we have found that there is no connection between the number of positive biomarkers and the severity of endoscopic activity when only considering the patients with endoscopic activity (*p* = 0.94), but this result was probably influenced by the low number of patients with different grades of severity (17 mild, 20 moderate and 5 severe).

Regarding the limitations of this study, it must be pointed out that the Cp cut-off (50 μg/g) was lower than the most commonly used thresholds in clinical practice for IBD monitoring [9,10], which has led to low specificity, PPV and AUROC values. Although the combination with the other three biomarkers provided a high diagnostic accuracy for the diagnosis of inflammatory activity, it can be hypothesized that increasing this cut-off or adding a quantitative Cp test could improve this result. This may lead to future investigation in this setting. Furthermore, we have included a limited number of participants (*n* = 106). Despite this fact, the test showed a noteworthy diagnostic accuracy with high AUROC, as well as high PPV and NPV values in patients with combined positivity or negativity of the four biomarkers. The low sample size probably conditioned the absence of connection between the number of positive biomarkers and the severity of the endoscopic activity. Another limitation was the lack of histological samples in all the patients, which appears to be the most objective evidence of absence of inflammatory activity and what best correlates with a good prognosis in IBD patients [29]. We solved that issue by accurately using the inflammatory endoscopic scales: Mayo, SES-CD, and Rutgeerts.

## 5. Conclusions

This test may be a useful strategy to monitor the inflammatory activity of IBD patients in clinical practice. The positivity or negativity of all biomarkers (detected in more than 30% of patients) yielded a high diagnostic accuracy, and the intermediate results provided acceptable NPV and PPV although these results did not correlate with the severity of the endoscopic activity. This test could be used at home by patients themselves reducing the number of outpatient visits and potentially improving their follow-up and consequently their quality of life.

## Figures and Tables

**Figure 1 diagnostics-11-02323-f001:**
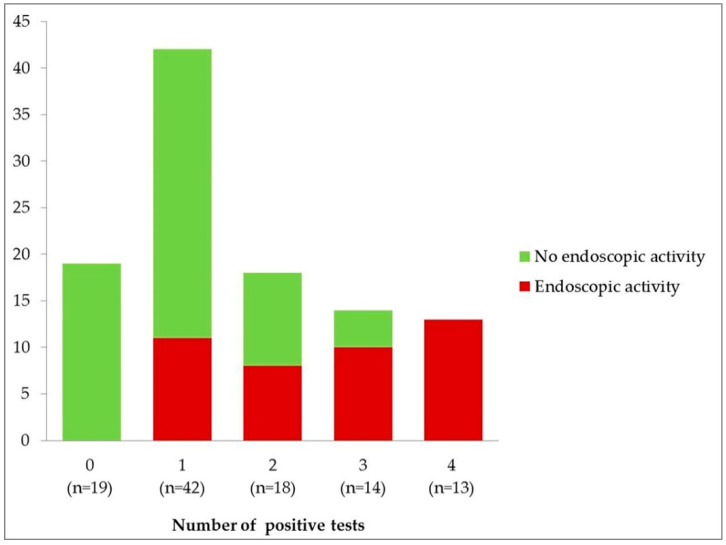
Number of patients with or without endoscopic activity according to the number of positive fecal tests.

**Figure 2 diagnostics-11-02323-f002:**
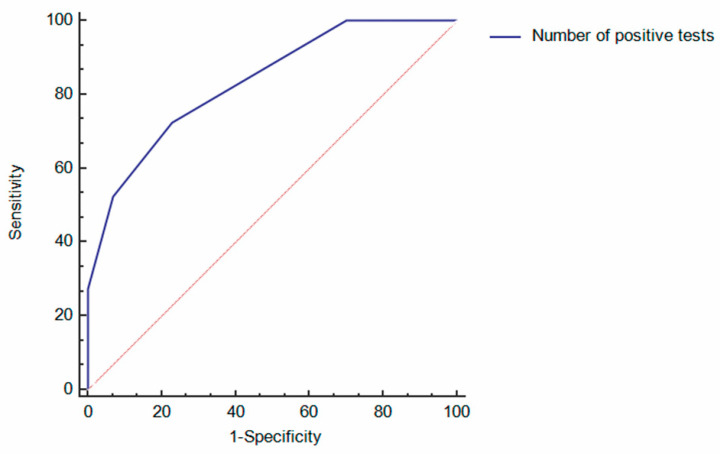
Area under receiver operation curve (AUROC) of the combo test for the diagnosis of inflammatory activity.

**Table 1 diagnostics-11-02323-t001:** Population baseline characteristics.

		IBD	UC	CD
Number of patients		106	54 (50.9%)	52 (49.1%)
Sex				
	Female	60 (56.7%)	31 (57.4%)	29 (55.7%)
	Male	46 (43.3%)	23 (42.6%)	23 (44.3%)
Age (years)		51.5 (SD 12.3)	52.5 (SD 11.4)	49.5 (SD 12.9)
Duration of disease (years)		14.4 (SD 9.1)	13 (SD 8.5)	16 (SD 9.6)
Smokers				
	No/past use	80 (75.4%)	47 (87.0%)	33 (63.4%)
	Current	26 (24.6%)	7 (13.0%)	19 (36.6%)

IBD: Inflammatory bowel disease, UC: Ulcerative colitis, CD: Crohn’s disease. SD: standard deviation.

**Table 2 diagnostics-11-02323-t002:** Results and diagnostic accuracy of the 4 independent biomarkers.

All Patients (*n* = 106)	Cp	Lf	Hb	Tf
True Positives	41	23	25	19
False Positives	42	5	8	8
True Negatives	22	59	56	56
False Negatives	1	19	17	23
Sensitivity (IC 95%)	97.6% (91.8–100)	54.8% (38.5–71)	59.5% (43.5–75.6%)	45.2% (29–61.5)
Specificity (IC 95%)	34.4% (22–46.8)	92.2% (84.8–99.5)	87.5% (78.6–96.4)	87.5% (78.6–96.4)
PPV (IC 95%)	49.4% (38–60.7)	82.1% (66.2–98.1)	75.8% (59.6–91.9)	70.4% (51.3–89.5)
NPV (IC 95%)	95.7% (85.1–100)	75.6% (65.5–85.8)	76.7% (66.3–87.1)	70.9% (60.2–81.5)

Cp: calprotectin; Lf: lactoferrin; Hb: haemoglobin; Tf: transferrin; PPV: positive predictive value; NPV: negative predictive value.

**Table 3 diagnostics-11-02323-t003:** Results and diagnostic accuracy of the 4 independent biomarkers stratified by diagnosis (Crohn’s disease and ulcerative colitis).

Ulcerative Colitis (*n* = 54)	Cp	Lf	Hb	Tf
True Positives	14	10	9	8
False Positives	27	1	5	6
True Negatives	12	38	34	33
False Negatives	1	5	6	7
Sensitivity (IC 95%)	93.3% (77.4–100)	66.7% (39.5–93.9)	60% (31.2–88.1)	53.3% (24.8–81.9)
Specificity (IC 95%)	30.8% (15–46.5)	97.4% (91.2–100)	87.2% (75.4–99)	84.6% (72–97.2)
PPV (IC 95%)	34.1% (18.4–49.9)	90.9% (69.4–100)	64.3% (35.6–93)	57.1% (27.6–86.6)
NPV (IC 95%)	92.3% (74–100)	88.4% (77.6–99.1)	85% (72.7–97.3)	82.5% (69.5–95.5)
**Crohn’s disease** **(*n* = 52)**	**Cp**	**Lf**	**Hb**	**Tf**
True Positives	27	13	16	11
False Positives	15	4	3	2
True Negatives	10	21	22	23
False Negatives	0	14	11	16
Sensitivity (IC 95%)	100% (98.1–100)	48.1% (27.4–68.8)	59.3% (38.9–79.6)	40.7% (20.4–61.1)
Specificity (IC 95%)	40% (18.8–61.2)	84% (67.6–100)	88% (73.3–100)	92% (79.4–100)
PPV (IC 95%)	64.3% (48.6–78)	76.5% (53.4–99.6)	84.2% (65.2–100)	84.6% (61.2–100)
NPV (IC 95%)	100% (95–100)	60% (42.3–77.6)	66.7% (49.1–84.3)	59% (42.2–75.7)

Cp: calprotectin; Lf: lactoferrin; Hb: haemoglobin; Tf: transferrin; PPV: positive predictive value; NPV: negative predictive value.

**Table 4 diagnostics-11-02323-t004:** Results and diagnostic accuracy of the combined rapid fecal test, considering the four possible cut-offs according to the number of positive tests.

All Patients (*n* = 106)	≥1 Test	≥2 Tests	≥3 Tests	4 Tests
True Positives	42	31	23	13
False Positives	45	14	4	0
True Negatives	19	50	60	64
False Negatives	0	11	19	29
Sensitivity (IC 95%)	100% (98.8–100)	73.8% (59.3–88.3)	54.8% (38.5–71)	30.9% (15.8–46.1)
Specificity (IC 95%)	29.7% (17.7–41.7)	78.1% (67.2–89)	93.7% (87.1–100)	100% (99.2–100)
PPV (IC 95%)	48.3% (37.2–59.3)	68.9% (54.2–83.5)	85.2% (69.9–100)	100% (96.1–100)
NPV (IC 95%)	100% (97.4–100)	82% (71.5–92.4)	75.9% (65.9–86)	68.8% (58.9–78.8)

PPV: positive predictive value; NPV: negative predictive value.

**Table 5 diagnostics-11-02323-t005:** Results and diagnostic accuracy of the combined rapid fecal test, considering the four possible cut-offs according to the number of positive tests, stratified by diagnosis (Crohn’s disease and ulcerative colitis).

Ulcerative Colitis (*n* = 54)	≥1 Test	≥2 Tests	≥3 Tests	4 Tests
True Positives	15	12	9	5
False Positives	29	9	2	0
True Negatives	10	30	37	39
False Negatives	0	3	6	10
Sensitivity (IC 95%)	100% (96.7–100)	80% (56.4–100)	60% (31.9–88.1)	33.3% (6.1–60.5)
Specificity (IC 95%)	25.6% (10.6–40.6)	76.9% (62.4–91.4)	94.9% (86.7–100)	100% (98.7–100)
PPV (IC 95%)	34.1% (19–49.2)	57.1% (33.6–80.7)	81.8% (54.5–100)	100% (90–100)
NPV (IC 95%)	100% (95–100)	90.9% (79.6–100)	86% (74.5–97.6)	79.6% (67.3–91.9)
**Crohn’s disease** **(*n* = 52)**	**≥1 test**	**≥2 tests**	**≥3 tests**	**4 tests**
True Positives	27	19	14	8
False Positives	16	5	2	0
True Negatives	9	20	23	25
False Negatives	0	8	13	19
Sensitivity (IC 95%)	100% (98.1–100)	70.4% (51.3–89.4)	51.9% (31.1–72.5)	29.6% (10.5–48.7)
Specificity (IC 95%)	36% (15.2–56.7)	80% (62.3–97.7)	92% (79.4–100)	100% (98–100)
PPV (IC 95%)	62.8% (47.2–78.4)	79.2% (60.8–97.5)	87.5% (68.2–100)	100% (93.7–100)
NPV (IC 95%)	100% (94.4–100)	71.4% (52.9–89.9)	63.9% (46.8–81)	56.8% (41–72.6)

PPV: positive predictive value; NPV: negative predictive value.

## Data Availability

The data that support the findings of this study are available on request from the corresponding author.

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
