# Peer review of "A Patient Self-Made Point-of-Care Fecal Test Improves Diagnostic Accuracy Compared with Fecal Calprotectin Alone in Inflammatory Bowel Disease Patients"

_diagnostics, 2021, doi:10.3390/diagnostics11122323_

Round 1
Reviewer 1 Report
Interesting study, well designed and written. A POCT reliable test definitely decreases the overall cost of healthcare and probably a positive POCT may even re inforce need for colonoscopy before guiding changes to IBD therapy in patient s who otherwise were not likely to follow up with colonoscopy based on recommendation for GI physician based off their symptoms alone.
The fecal calprotectin cut off of 50 is too low in clinical practice to consider to corelate with endoscopic activity . Mostly 50-120 is borderline elevated and recommended to re evaluate in 4-6 week- ish time due to chances of elevation being false positive , > 120 is strongly positive and corelates more so with endoscopic activity in IBD. Did the very low cut off used as positive influence the fact the specificity of calprotectin in your analysis? and if so , is there anyway to look at data with higher positive cut off for calprotectin to see if the results change?
Also, was there any analysis done to look at data of positive results on combination of calprotectin and lactoferrin ( knowing these are inflammatory markers while FOB and transferrin are not so) and compare it to combination of all 4 test returning positive with respect to co relation with colonoscopic activity ?
Changes in writing style recommended: "Colonoscopy continues being the gold standard for this goal, but it is not exempt from complications, highly economic cost, and sometimes poor acceptance by patients. Nowadays, we are living in a difficult sanitary situation due to COVID-19 pandemic, which has complicated the monitoring of all chronic diseases including IBD patients".
Reviewer 2 Report
This is an interesting paper that reported the 4-biomarker combination fecal test as a useful tool for diagnosis of IBD (UC and BD) inflammatory activity. The manuscript is generally well written but needs English copyediting and reconsideration for the following points.
- Line 20: "For a single-center,,,, consectuive,,,,,,,,,."?
- Line 24/25: "ulcerative colitis (not Ulcerative Colitis)", "Crohn's disease (not Crohn's Disease)".
- Line 28: "lesser (than calprotectin) specificity"?
- Line 33: Remove "separately". "This 4 biomarker (combination) point of care test" would be better.
- Line 40: "ulcerative colitis (UC)"
- Line 43; I don't understand what "some patients had recurrent and resistant to treatment disease" means.
- There are too many paragraphs in the Introduction and Discussion, making it somewhat hard to read. Combine some of them!
- Line 105/106: g rather gr for gram.
- Line 129: Remove "separately"; "and their combination,".
- Line 132/133: Add names of software companies.
- Line 135: Add approval number and date.
- Table 2/3 legends: ",,,accuracy by the 4 independent biomarkers,,,," ; remove "separately"
- Table 2: Tf false negatives " 7+16=23 but not 19".
- Line 158/159: Where (any table?) can we find out that "A 17.9% (19/106),,,,,, in the four biomarkers"?
- Line 214: Remove "separately"
- Line 241: "a low PPV (49.3%)" or 49.4% (in Table 2)?
- Line 255: "92.1%" or 92.2% (in Table 2)?
- Line 260: What do the authors mean by "better" results?
- Line 267: Add (Table 4) after PPV. Similarly, insert referred Table/Fig No. for each comment in the Discussion.
Round 2
Reviewer 1 Report
no further comments